# A View From Somewhere: Human-Centric Face Representations

**Jerone T. A. Andrews**[*]
Sony AI, Tokyo

**Przemysław Joniak**[†]
University of Tokyo, Tokyo

**Alice Xiang**
Sony AI, New York

## Abstract

We propose to implicitly learn a set of continuous face-varying dimensions, without ever asking an annotator to explicitly categorize a person. We uncover the dimensions by learning on a novel dataset of 638,180 human judgments of face similarity (FAX). We demonstrate the utility of our learned embedding space for predicting face similarity judgments, collecting continuous face attribute values, and attribute classification. Moreover, using a novel conditional framework, we show that an annotator's demographics influences the importance they place on different attributes when judging similarity, underscoring the need for diverse annotator groups to avoid biases.

## 1 Introduction

Most human-centric image datasets are web-scraped, lacking ground-truth information about the data subjects. Moreover, data protection legislation considers demographic attributes to be personal information and limits their collection and use [4, 3]. Even when labels are known, evaluating diversity by examining counts across subgroups fails to reflect the continuous nature of human phenotypic diversity (e.g., skin tone is often reduced to `light` vs. `dark`). Furthermore, such an approach often denies multi-group membership [60] (e.g., erasing multi-ethnic and intersex individuals).

When labels are unknown, researchers typically choose certain attributes they consider to be *relevant* for human diversity and use human annotators to infer them [32, 66, 77]. In practice, this is difficult for ill-defined and highly changeable social constructs such as race and gender [22, 34, 9]. Observational labels from annotators risk not only encoding *stereotypes*, but reifying and propagating them beyond "their cultural context" [35, 18]. Furthermore, discrepancies between, e.g., observed and self-identified race, gender, or other sensitive attributes can induce psychological distress [11] by invalidating an individual's self-image and identity [59].

**Contributions.** As an image subject can exhibit certain traits to a greater or lesser extent than others, even within the same subpopulation, the goal of our work is to learn a *similarity function*, which measures the similarity between two faces in a way that is *aligned with human perception*. We aim to do so without using categorical labels.

Concretely, we present a model of conditional decision-making for learning continuous, human-interpretable face embeddings directly from human judgments of face similarity. Underlying our approach is FAX, a novel dataset of 638,180 judgments over 4,921 faces, where each judgment corresponds to the odd-one-out (i.e., least similar) face in a triplet of faces. Such *contextual* similarity judgments have been shown to offer a *window* into the dimensions (i.e., factors of variation) of object categories [27, 84] and reachspace environments [31] in the human mind. Distinct from other computer vision datasets, each judgment in FAX is accompanied by the identifier and demographic attributes of the annotator who made the judgment.

---

[*]Corresponding author: `jerone.andrews@sony.com`.
[†]Work done while author was an intern at Sony AI, Tokyo.

2022 Trustworthy and Socially Responsible Machine Learning (TSRML 2022) co-located with NeurIPS 2022.

Fig. 1 illustrates our method, which concurrently learns an embedding space and the *importance* distinct annotators place on each of the embedding dimensions when determining similarity. By learning importance scores, we induce subspaces that encode each annotator's notion of similarity. We

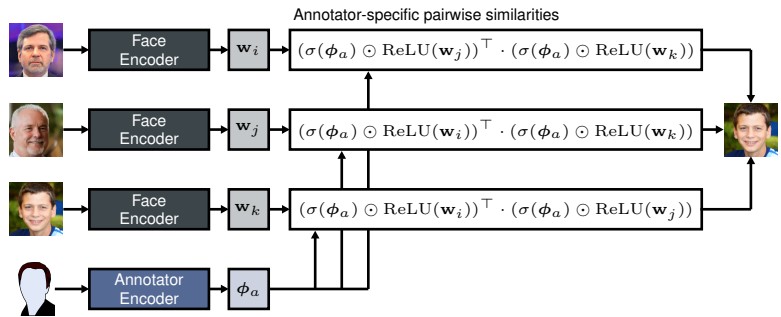

Figure 1: Model of Conditional Decision-Making.

show that predicting similarity judgments from an annotator in their respective subspace increases predictive accuracy. Significantly, we find that annotator subspaces are not interchangeable, but can be *grouped* wrt the sociocultural backgrounds (e.g., nationality, ancestry) of the annotators. Our results underscore the need for diverse annotator groups to avoid biases.

Additionally, we show that our embeddings are highly correlated with the human mental representational space of faces, as compared to face embeddings induced by learning to predict face identity or face attributes. Importantly, we find that the individual embedding dimensions are: related to concepts of gender, race, age, as well as face and hair morphology. We demonstrate the utility of such dimensions for: (i) collecting continuous (as opposed to discrete) face attribute values for novel faces from annotators; and, (ii) binary attribute classification.

## 1.1 Background

**Disentangled Representation Learning.** Learning interpretable representations through semantically labeled concepts and prototypes has recently gained attention [2, 38, 43]. Others reveal meaningful directions inside generative models, where directional distances estimate the magnitude of attributes [53]. However, the identification of interpretable directions is performed post-hoc, requiring potentially biased labeled examples and/or pretrained models [81, 47], self-supervision [57, 30], or paired images sharing at least one factor of variation [48]. Our approach is similar in spirit to wholly unsupervised approaches [23, 55, 68, 79]. Rather than modeling the data generating distribution, our representations are trained to predict human similarity judgments, reflecting the human mental representational space of faces. Our baselines primarily center on embeddings induced by training on facial analysis datasets, since in the face domain, post-hoc interpretation of latent dimensions [69, 53, 81, 47] predominantly rely on datasets such as CelebA [46].

**Psychological Embeddings.** Multidimensional scaling (MDS) is often used to learn psychological embeddings from pairwise similarities [72, 84, 27, 58, 73, 31]. Zheng et al. [84] developed an MDS approach for learning psychological object embeddings from odd-one-out judgments based on three assumptions. First, embeddings can be learned solely from odd-one-judgments, where representations are constrained to be continuous, non-negative, and sparse. Such properties support interpretability such that dimensions indicate both feature presence and feature magnitude. Second, odd-one-out judgments are a function of $\text{sim}(i,j)$, $\text{sim}(i,k)$, and $\text{sim}(j,k)$, where $\mathbf{x} \in \mathcal{X}$ and $\text{sim} : (\mathbf{x}_i, \mathbf{x}_j) \rightarrow \text{sim}(i,j) \in \mathbb{R}$ denote a face image and similarity function, resp. Third, odd-one-out judgments are stochastic, where the probability of selecting $x_k$ as the odd-one-out is $p(k) \propto \exp(\text{sim}(i,j))$. As MDS cannot embed data outside of the training set, researchers have used pretrained models to predict MDS representations [64] or directly infer similarity judgments [56, 6]. Rather than relying on pretrained models which can introduce implicit biases [39, 71], we perform end-to-end learning from scratch. Further, unlike recent work [27, 31], we focus on human perception of naturalistic face images, evidenced to be processed differently to objects in the human visual system [70].

**Annotator Bias.** Categorization by observation depends not only on the subject being categorized, but also on the annotator's sociocultural background, perception of the subject, and contextual cues [67, 7, 29, 59]. For instance, a subject's clothing can influence their racial categorization [19]. Despite this, annotator positionality has only recently entered into discourse in computer vision [16, 83, 65], albeit predominantly in the context of natural language processing [1, 78, 80, 10, 62]. However, in order to mitigate bias, one must first measure bias [41]. Unlike previous works, our model of conditional

decision-making elucidates the *importance* distinct annotators place on different attributes when judging similarity, thus permitting the mitigation of bias. Veit et al. [74] employ a similar procedure to learn subspaces which encode different notions of similarity (e.g., font style, character type).

## 2 Method

To learn a similarity function aligned with human perception, we collect a large-scale dataset, FAX, of odd-one-out similarity judgments. An odd-one-out judgment corresponds to the least similar face in a triplet of face images, representing a *three alternative forced choice task*.

**Data.** The benefits of odd-one-out judgments are threefold. Most significantly, the judgments do not require an annotator to categorize people. Thus, we do not encode, reify, or propagate stereotypes. Second, for a triplet of faces $(\mathbf{x}_i, \mathbf{x}_j, \mathbf{x}_k)$, repeatedly varying $\mathbf{x}_k$ permits the identification of the *relevant* dimensions that contribute to $\text{sim}(i, j)$. That is, w.l.o.g., $\mathbf{x}_k$ provides context for which $\text{sim}(i, j)$ is determined, making the task easier than *explicit* pairwise similarity tasks (i.e., "Is $\mathbf{x}_i$ similar to $\mathbf{x}_j$?"). This is because it is not always apparent to an annotator which dimensions are relevant when determining $\text{sim}(i, j)$, especially when $\mathbf{x}_i$ and $\mathbf{x}_j$ are perceptually different. Finally, there is no need to prespecify attribute lists hypothesized as relevant for comparison (e.g., "Is $\mathbf{x}_i$ *older* than $\mathbf{x}_j$?"). The odd-one-task instead implicitly uncovers salient attributes.

For our proof of concept, the 4,921 faces were sampled from the CC-BY licensed dataset FFHQ [33]. The subset was obtained by first splitting FFHQ into 56 partitions based on estimated intersectional group labels, and then randomly sampling from each partition with equal probability. All faces are near-frontal with little to no eye occlusions and an apparent age $> 19$ years old.

FAX contains 638,180 quality-controlled triplets over the 4,921 faces, representing 0.003% of all possible triplets. Each triplet is labeled with an odd-one-out judgment and the identifier of the annotator who made the judgment. As in previous non-facial odd-one-out datasets [31, 28], there is a single judgment per triplet. In total, 1,645 annotators contributed to FAX over Amazon Mechanical Turk (AMT), providing their consent, age, nationality, ancestry, and gender identity. Compensation was 15 USD per hour. Additional details regarding FAX are provided in Appendix A

**Model of Conditional Decision-Making.** At a high-level, we want to apply Zheng et al.'s MDS approach (Section 1.1) to learn face embeddings. However, MDS cannot embed images outside of the training set, limiting its utility. Moreover, MDS pools all judgments, disregarding intra- and inter-annotator stochasticity. To resolve these issues, we propose to learn a *conditional* convolutional neural network (CNN).

Let $\{(\{\mathbf{x}_{i\ell}, \mathbf{x}_{j\ell}, \mathbf{x}_{k\ell}\}, k\ell, a)\}_{\ell=1}^n$ denote a training set of $n$ (triplet, judgment, annotator) tuples, where $a \in \mathcal{A}$. To simplify notation, we assume that judgments always correspond to the index $k\ell$. Suppose $f : \mathbf{x} \mapsto f(\mathbf{x}) = \mathbf{w} \in \mathbb{R}^d$ is a CNN, parametrized by $\Theta$, where $\mathbf{w} \in \mathbb{R}^d$ is an embedding of $\mathbf{x}$. We model the probability of annotator $a$ selecting $k\ell$ as $p(k\ell \mid a) \propto \exp(\text{sim}_a(i\ell, j\ell))$, where $\text{sim}_a(\cdot, \cdot)$ denotes the *internal* similarity function of $a$. Given two images $\mathbf{x}_i$ and $\mathbf{x}_j$, we define their similarity according to $a$ as: $\text{sim}_a(i, j) = (\sigma(\boldsymbol{\phi}_a) \odot \text{ReLU}(\mathbf{w}_i))^\top \cdot (\sigma(\boldsymbol{\phi}_a) \odot \text{ReLU}(\mathbf{w}_j))$, where $\sigma(\cdot)$ and $\text{ReLU}(\cdot)$ denote the sigmoid and ReLU functions, resp. Note that $\sigma(\boldsymbol{\phi}_a) \in [0, 1]^d$ is a *mask* associated with $a$. Each mask plays the role of an element-wise gating function, encoding the *importance* $a$ places on each of the possible $d$ embedding dimensions when determining similarity. By conditioning prediction on annotator identifiers, we induce subspaces that encode each annotator's notion of similarity. This allows us to study whether per-dimensional importance scores differ between annotators.

Let $\boldsymbol{\Phi} = [\boldsymbol{\phi}_1^\top, \ldots, \boldsymbol{\phi}_{|\mathcal{A}|}^\top] \in \mathbb{R}^{d \times |\mathcal{A}|}$ denote a trainable weight matrix, where each column vector $\boldsymbol{\phi}_a^\top$ corresponds to annotator $a$'s mask prior to applying $\sigma(\cdot)$. In our conditional framework, we jointly optimize $\Theta$ and $\boldsymbol{\Phi}$ by minimizing:

$$-\sum_\ell \log\left[\hat{p}(k\ell \mid a)\right] + \lambda_1 \sum_i \|\text{ReLU}(\mathbf{w}_i)\|_1 + \lambda_2 \sum_{ij} \text{ReLU}(-\mathbf{w}_i)_j + \lambda_3 \sum_a \|\boldsymbol{\phi}_a\|_2^2, \quad (1)$$

where $\hat{p}(k\ell \mid a) = \frac{\exp(\text{sim}_a(i\ell, j\ell))}{\exp(\text{sim}_a(i\ell, j\ell)) + \exp(\text{sim}_a(i\ell, k\ell)) + \exp(\text{sim}_a(j\ell, k\ell))}$ is the predicted probability that $k\ell$ is the odd-one-out conditioned on $a$. The first term in Eq. 1 encourages similar face pairs to result in large dot products. The second term (modulated by $\lambda_1 \in \mathbb{R}$) promotes sparsity. The third

term (modulated by $\lambda_2 \in \mathbb{R}$) supports interpretability by penalizing negative values. The last term (modulated by $\lambda_3 \in \mathbb{R}$) penalizes large weights.

When $\lambda_3 = 0$ and $\sigma(\phi_a) = [1, \dots, 1]$ ($\forall a$), Eq. 1 corresponds to the *unconditional* MDS objective proposed by Zheng et al. [84]. In what follows, we refer to unconditional and conditional models trained on FAX as `FAX-U` and `FAX-C`, resp. In addition, we refer to conditional models whose masks are learned post hoc as `FAX-C-PH`. A `FAX-C-PH` model uses a fixed unconditional model (i.e., `FAX-U`) to obtain face embeddings such that only $\mathbf{\Phi}$ is trainable.

Significantly, our conditional decision-making framework is generally applicable to any task that involves mapping from inputs to human decisions; it only requires record-keeping during data collection such that each judgment (or annotation) is associated with the annotator who generated it.

**Implementation.** All models have ResNet18 [25] architectures and output 128-dimensional embeddings. We use the Adam [37] optimizer with default parameters, reserving 10% of FAX for validation. Based on grid search, we empirically set $\lambda_1 = 5 \times 10^{-5}$ and $\lambda_2 = 1 \times 10^{-2}$. For `FAX-C` and `FAX-C-PH`, we additionally set $\lambda_3 = 1 \times 10^{-5}$. Across independent runs, we find that: (i) only a fraction of the 128 dimensions are needed; and (ii) individual dimensions are reproducible. Post-optimization, we remove dimensions with maximal values close to zero. For `FAX-U`, this results in 22 dimensions (61.9% validation accuracy). Further details are in Appendix B.

## 3 Experiments

We demonstrate the utility of FAX induced embedding spaces for predicting face similarity judgments, revealing annotator bias, collecting continuous face attribute values, and attribute classification. Throughout this paper, statistical tests correspond to Mann-Whitney $U$-tests ($p$-value $< 0.05$).

For comparison, we consider embeddings extracted from pretrained face recognition and facial analysis models. In particular, an ArcFace [17] face recognition model trained on CASIA-Webface (CWF) [82]; and analysis models trained on CelebA (CA) [46] (40 binary face attribute labels), Fair-Face (FF) [32] (perceived binary gender expression, age, and race labels), and FFHQ (binary gender expression, age, and race labels). All models have ResNet18 architectures and output 128-dimensional embeddings. For the analysis models, we also consider unnormalized class logit representations, as well as the logit representations converted to one-hot encodings. All representations are normalized to unit-length, and the dot product of two representations determines their similarity. Additional details and experimental results are in Appendix C.

**Predicting Human Similarity Judgments.** The utility of any face embedding method is typically determined by measuring whether similar faces are *closer* together in feature space than dissimilar faces. We therefore employ two evaluation protocols, evaluating odd-one-out predictive accuracy.

In **Protocol 1**, we test whether our embeddings are predictive of human judgments not observed during learning. This is a typical MDS test setting, as MDS cannot embed images outside of the training set. Using images from the stimulus set of 4,921 faces, we generate 1,000 novel triplets and collect 22–25 unique judgments on AMT per triplet (24,060 judgments). In addition to predictive accuracy, we compute Spearman's $r$ correlation between the entropy in human- and model-generated triplet odd-one-out probabilities. Human-generated odd-one-out probabilities are of the form: $(n_i, n_j, n_k)/n$, where, w.l.o.g., $n_k/n$ corresponds to the fraction of $n$ odd-one-out votes for $k$.

As we have 22–25 judgments per triplet in **Protocol 1**, we can reliably estimate odd-one-out probabilities. The Bayes optimal classifier accuracy corresponds to the *best* possible accuracy any model could achieve given the stochasticity in the human judgments. The classifier makes the most probable prediction, i.e., the majority judgment. Its accuracy is equal to the mean majority judgment probability over the 1,000 triplets, corresponding to 65.5±1% (95% CI).

In **Protocol 2**, we sample 56 novel face images from FFHQ not contained in the stimulus set. We then generate all $\binom{56}{3}$ possible triplets and collect 2–3 unique judgments on AMT per triplet (80,300 judgments). As an additional performance measure, we compute Spearman's $r$ between the strictly upper triangular model- and human-generated similarity matrices. Entry $(i, j)$ in the human-generated similarity matrix corresponds to the fraction of triplets containing $(i, j)$, where neither was judged as the odd-one-out. Entry $(i, j)$ in a model-generated similarity matrix corresponds to the mean $\hat{p}(i, j)$ over all triplets containing $(i, j)$.

Results for **Protocol 1** and **Protocol 2** are shown in Tbl. 1. First, embeddings trained on FAX outperform the baselines across all metrics. In particular, as evidenced by the correlation tests, the human mental representational space is better represented by embedding spaces induced by learning on FAX. This even holds for our `FAX-U` model trained on *only* 72K judgments. Second, `FAX-C` and `FAX-C-PH` have increased performance over their unconditional counterparts. This shows that the learned annotator-specific masks generalize, i.e., the encoded importance an annotator places on each dimension extends to novel judgments. Most significantly, in **Protocol 1**, our conditional models attain a predictive accuracy at or above the upper bound of the Bayes optimal classifier.

Table 1: (**Protocol 1**) Accuracy over 24,060 judgments and Spearman's $r$ between the entropy in human- and model-generated triplet odd-one-out probabilities. (**Protocol 2**) Accuracy over 80,300 judgments and Spearman's $r$ between the entropy in human- and model-generated similarity matrices.

| Model | Loss/Method | Protocol 1 | | Protocol 2 | |
|---|---|---|---|---|---|
| | | Acc. | $r$ | Acc. | $r$ |
| FF | Cross-entropy | 55.9 | 0.41 | 51.9 | 0.67 |
| CA | Cross-entropy | 52.1 | 0.25 | 48.9 | 0.56 |
| FFHQ | Cross-entropy | 55.2 | 0.38 | 51.8 | 0.68 |
| CWF | ArcFace | 51.6 | 0.29 | 46.1 | 0.40 |
| FAX-C | Conditional eq. (1) | *67.4* | *0.68* | *61.7* | 0.82 |
| FAX-C-PH | Conditional eq. (1) | 66.5 | *0.68* | 61.4 | 0.82 |
| FAX-U | Unconditional eq. (1) | 62.0 | 0.65 | 57.5 | *0.86* |
| FAX-U·1/2 | Unconditional eq. (1) | 61.3 | 0.61 | 55.8 | 0.81 |
| FAX-U·1/4 | Unconditional eq. (1) | 60.6 | 0.56 | 55.3 | 0.80 |
| FAX-U·1/8 | Unconditional eq. (1) | 58.6 | 0.47 | 55.0 | 0.79 |

**Annotator Bias.** Conditioning prediction on annotator identifiers provides the best predictive accuracy, evidencing that knowledge of the annotator determining similarity assists in informing the outcome. However, annotators are often framed as interchangeable [51, 14]. To test the validity of this assumption, in **Protocol 2**, we randomly swap the annotator associated with each judgment and then recompute the predictive accuracy of `FAX-C`. Repeating this process 100 times results in a performance drop from 61.7% to 52.8% $\pm$ 0.02% (95% CI) on average. This shows that annotator subspaces are not interchangeable.

An interesting question relates to whether an annotator's sociocultural background influences their decision making. To evaluate this, we create sets $\{(\sigma(v_a), y)\}$, where $\sigma(v_a)$ and $y \in \mathcal{Y}$ are annotator $a$'s learned mask and self-identified demographic attribute, resp. For a particular annotator attribute (e.g., nationality), we limit the dataset to annotators who contributed $\geq 200$ judgments and belong to one of the two largest groups wrt the attribute. Using 10-fold cross validation, we

Table 2: Results for linear SVMs trained to discriminate between binary annotator demographic groups.

| Annotator Groups | #Masks | AUC |
|---|---|---|
| 30-39 / 40-49 | 393 / 121 | 0.59 $\pm$ 0.05 |
| Male / Female | 523 / 473 | 0.65 $\pm$ 0.05 |
| American / Indian | 530 / 204 | 0.86 $\pm$ 0.03 |
| European / Asian | 407 / 243 | 0.86 $\pm$ 0.03 |
| West European / South Asian | 173 / 107 | 0.88 $\pm$ 0.05 |

train linear support vector machines (SVMs) [26] with balanced class weights to predict $y$ from $\sigma(v_a)$. Tbl. 2 shows the average area under the receiver operating characteristic (AUC) for each attribute. Most significantly, none of the AUC confidence intervals include chance performance. Moreover, the linear SVMs are able to discriminate between binary groups wrt nationality, regional ancestry, and subregional ancestry with high probability (86–88%).

**Interpretability.** Since we aim to replace the *explicit* collection of problematic categorical labels, we evaluate whether the individual `FAX-U` (and `FAX-C-PH`) dimensions are human-interpretable through a **(qualitative) dimension labeling task** and a **(quantitative) dimension rating task** [84, 27, 31].

For the **dimension labeling task**, we task annotators with writing descriptions about each of the 22 dimensions using continuous *dimensional scales*. A dimensional scale corresponds to the the stimulus set sorted in descending order based on their values in the dimension. Consistent descriptions evidence that a dimension is meaningful. We collect 25–62 descriptions per dimension. Based on the descriptions, we observe distinct dimensions coinciding with commonly defined demographic attribute concepts, i.e., `Male`, `Female`, `Black`, `White`, `East Asian`, `South Asian`, and `Elderly`. In addition, there are separate dimensions for face and hair morphology, i.e., `Wide Face`, `Long Face`, `Smiling Expression`, `Neutral Expression`, `Balding`, `Facial Hair`, and `Dyed Hair`. (See Fig. 2.)

For the **dimension rating task**, we task annotators with placing 20 novel faces on each of the 22 dimensional scales. We collect 20 unique judgments per dimension for each face. Separately for each face, we average the judgments per dimension such that we obtain human-generated embeddings. Using the human-generated embeddings, we create a human-generated similarity matrix and compare

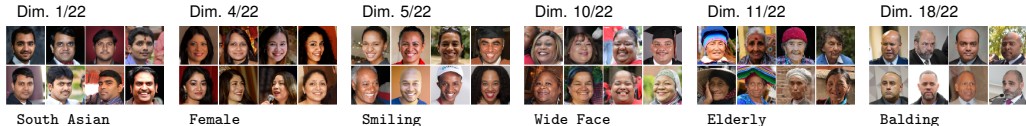

| Dim. 1/22 | Dim. 4/22 | Dim. 5/22 | Dim. 10/22 | Dim. 11/22 | Dim. 18/22 |
|:---:|:---:|:---:|:---:|:---:|:---:|
| South Asian | Female | Smiling | Wide Face | Elderly | Balding |

**Dimension Labeling Task.** We show a subset of the 22 `FAX-U` (and `FAX-C-PH`) dimensions. Each dimension is shown with the top 8 faces from the stimulus set with the highest dimensional values.

Figure 2: Dimension labeling task.

it to a model-generated similarity matrix. The model-generated similarity matrix is created using model-generated embeddings. Spearman's $r$ correlation between the strictly upper triangular model- and human-generated similarity matrices is 0.83 and 0.86 for `FAX-U` and `FAX-C-PH` embeddings, resp. This shows that: (i) dimensional values correspond to the feature magnitude; and (ii) dimensional scales can be used to *directly* collect continuous attribute values for faces, sidestepping the limits of categorical definitions [34, 9, 35]. Note that the use of a dimensional scale is not restricted to any particular embedding method, as long as the dimension is human-interpretable.

**Binary Attribute Classification.** Facial analysis models are trained for classification, therefore we test the utility of `FAX-U` dimensions for discriminating between binary-valued attributes. For labeled face data, we use COCO [44], OpenImages MIAP (MIAP) [66], CFD, FFHQ, CelebA (CA) [46], and Casual Conversations (CC) [24]. Tbl. 3 shows that `FAX-U` dimensions are competitive with the baselines, even in challenging unconstrained settings as represented by COCO and MIAP. Therefore, training on FAX results in interpretable dimensions that can serve as classifiers. Notably, `FAX-U` dimensions have the highest AUC for self-identified CFD attributes `Black`, `East Asian`, and `Indian`.

Table 3: Using attribute values from relevant model dimensions to perform binary attribute classification.

| | | AUC | | | |
|---|---|---|---|---|---|
| Dataset | Attribute | FAX-U | CA | FF | FFHQ |
| CC | > 70 y.o.* | 0.800 | 0.936 | 0.959 | 0.962 |
| FFHQ | > 70 y.o. | 0.905 | 0.959 | *0.971* | *0.980* |
| CFD | Male* | 0.991 | 0.997 | 0.996 | *0.998* |
| CC | Male* | 0.971 | 0.986 | *0.990* | 0.981 |
| CA | Male | 0.990 | *0.999* | 0.994 | *0.995* |
| COCO | Male | 0.893 | 0.926 | *0.963* | 0.958 |
| MIAP | Male | 0.924 | 0.942 | *0.945* | 0.938 |
| FFHQ | Male | 0.933 | 0.959 | 0.988 | *0.996* |
| CA | Smiling | *0.895* | *0.982* | — | — |
| CFD | Open mouth | 0.969 | 0.992 | — | — |
| CFD | Neutral | *0.731* | — | — | — |
| CFD | East Asian* | *0.969* | — | 0.955 | 0.938 |
| CFD | Black* | *0.992* | — | 0.992 | 0.988 |
| CFD | White* | 0.972 | 0.889 | 0.990 | 0.988 |
| CFD | Indian* | *0.960* | — | 0.848 | 0.883 |
| CC | Light skin | 0.930 | 0.830 | *0.965* | 0.960 |
| COCO | Light skin | 0.889 | 0.771 | 0.927 | *0.931* |
| CA | Balding | *0.963* | *0.995* | — | — |

## 4 Conclusion

Motivated by issues inherent to categorization by observation, we proposed a method for implicitly learning continuous face-varying dimensions, without ever asking an annotator to explicitly categorize a person. We uncovered the face embedding space by learning on a novel dataset of human judgments of face similarity (FAX). We showed that the individual dimensions are human-interpretable and related to concepts of gender, race, age, as well as face and hair morphology categories. We demonstrated the utility of our learned embedding space for predicting face similarity judgments, collecting continuous face attribute values, and attribute classification. Moreover, using our novel conditional framework, we showed that an annotator's demographics influences the importance they place on different attributes when judging similarity, underscoring the need for diverse annotator groups to avoid biases. We hope that our work inspires others to pursue unorthodox tasks for learning face-varying dimensions, which do not encode, reify, and propagate stereotypes, or invalidate the self-image and identity of image subjects.

**Future Work.** As our dataset represents 0.003% of all possible triplets that can be sampled from 4,921 images. To go one step further, triplets should be sampled selectively so as to learn additional face attribute dimensions. That is, FAX can be extended using active learning approaches, e.g., focusing on triplets composed of faces that are close in embedding space whose similarity judgments are unlikely to be based on gender, skin color, or race, but rather finer-grained attributes such as hair color, eye color, nose shape, etc.

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

## A    Data

### A.1    Face image stimuli

The stimulus set was sampled from the publicly available FFHQ dataset [33] of 70,000 permissively licensed face images. The FFHQ dataset which includes JSON metadata and documentation is available from the following repository: https://github.com/NVlabs/ffhq-dataset.

For face detection, as in [33], we used the dlib mmod human face detector [36] (https://github.com/davisking/dlib-models). To align the detected faces, we followed the same procedure as detailed in [54], which uses the dlib 68 landmark shape predictor [36] (https://github.com/davisking/dlib-models).

As stated in the main text, we used FFHQ-Aging annotations [54], obtained using Face++, for absolute head yaw angle, absolute head pitch angle, eyewear, and eye occlusion annotations to prune the dataset. The purpose of these constraints was to reduce the influence of head pose on human behavior and to better allow for eye comparisons. The stimulus set of 4,921 representative examples were then sampled from the remaining images such that each intersectional subgroup based on perceived binary gender expression, age group, and ethnicity would be included. We only considered face images with an apparent age greater than 19 years old in an attempt to exclude images of minors. Ethnicity was estimated using a FairFace [32] trained model (implementation details are in Appendix B.2). Perceived binary gender expressions and apparent age groups were obtained from FFHQ-Aging crowdsourced annotations. To mitigate biasing our set toward stereotypical faces, we randomly sampled images from the 56 possible intersectional subgroups rather than selecting the most confidently predicted or annotated.

Table 4: Apparent age group, binary gender expression group, and ethnicity group counts for the stimuli of 4,921 representative examples in the face image stimuli set

| Age | Count |
|---|---|
| 20–29 | 1400 |
| 30–39 | 1351 |
| 40–49 | 1181 |
| 50–120 | 989 |

| Ethnicity | Count |
|---|---|
| White | 998 |
| East Asian | 801 |
| Latino Hispanic | 788 |
| Black | 729 |
| Indian | 556 |
| Middle Eastern | 538 |
| Southeast Asian | 511 |

| Gender | Count |
|---|---|
| Male | 2607 |
| Female | 2314 |

Table 4 shows the group counts at the level of age, gender, and ethnicity, whereas Table 6 reports the counts at the finer intersectional subgroup level. Exact balancing was not possible due to the skewed composition of FFHQ. Ideally, we would have sampled images from a dataset with self-reported attributes. However, while such datasets do exist, they are typically small, tightly constrained, and largely unrepresentative.

### A.2    Collection of human behavioral judgments

All participants were recruited on AMT and were required to have previously completed a minimum of 100 human intelligence tasks (HITs) on AMT with a 95% approval rating. Eligibility was further determined through a prescreening survey (nominal fee of 0.01 USD), which included a short multiple-choice English language proficiency test. Since we placed no restriction on participant location, in order to be eligible, we required participants to answer at least two out of three multiple-choice English language proficiency questions correctly.

**Collection of human behavioral odd-one-out similarity judgments for training and validation.**
Based on estimated face image labels, a triplet can comprise 1–2 binary gender expression group(s), 1–3 age group(s), and 1–3 ethnicity group(s) resulting in 18 triplet *types* which were uniformly sampled. We sampled 703,300 unique triplets in total. Participants were presented with triplets alongside the following minimal instructions: "Choose the person that looks *least* similar to the two other people.

Table 5: Participant exclusion policies and criteria

| Policy | Criteria | Min. # of judgments |
|---|---|---|
| Fast #1 | $> 25\,\%$ of judgments generated in $< 0.8\,$s | 100 |
| Fast #2 | $> 50\,\%$ of judgments generated in $< 1.1\,$s | 100 |
| Deterministic | $> 40\,\%$ of judgments correspond to a single triplet position | 200 |
| Incomplete | Submission of an empty judgment | 1 |

Table 6: The 56 intersectional subgroup counts for the stimuli of 4,921 representative examples

| Age | Ethnicity | Gender | Count |
|---|---|---|---|
| 50–120 | White | Male | 140 |
| 20–29 | White | Female | 135 |
| 40–49 | White | Male | 133 |
| 20–29 | White | Male | 127 |
| 30–39 | Latino Hispanic | Female | 126 |
| 30–39 | White | Male | 125 |
| 50–120 | White | Female | 118 |
| 20–29 | East Asian | Female | 118 |
| 30–39 | East Asian | Female | 115 |
| 30–39 | White | Female | 112 |
| 30–39 | East Asian | Male | 111 |
| 40–49 | White | Female | 108 |
| 20–29 | East Asian | Male | 107 |
| 20–29 | Latino Hispanic | Male | 106 |
| 40–49 | Black | Male | 105 |
| 40–49 | Latino Hispanic | Female | 105 |
| 20–29 | Latino Hispanic | Female | 104 |
| 30–39 | Black | Male | 104 |
| 40–49 | East Asian | Female | 102 |
| 30–39 | Middle Eastern | Male | 100 |
| 20–29 | Black | Male | 99 |
| 40–49 | Latino Hispanic | Male | 98 |
| 50–120 | Middle Eastern | Male | 97 |
| 40–49 | Middle Eastern | Male | 97 |
| 20–29 | Middle Eastern | Male | 96 |
| 30–39 | Black | Female | 95 |
| 20–29 | Indian | Female | 95 |
| 30–39 | Indian | Female | 94 |
| 40–49 | Black | Female | 93 |
| 30–39 | Latino Hispanic | Male | 93 |
| 20–29 | Black | Female | 92 |
| 40–49 | East Asian | Male | 91 |
| 30–39 | Indian | Male | 91 |
| 50–120 | East Asian | Male | 88 |
| 50–120 | Black | Male | 86 |
| 20–29 | Southeast Asian | Female | 86 |
| 20–29 | Indian | Male | 83 |
| 20–29 | Southeast Asian | Male | 82 |
| 50–120 | Latino Hispanic | Female | 81 |
| 50–120 | Latino Hispanic | Male | 75 |
| 30–39 | Southeast Asian | Male | 75 |
| 30–39 | Southeast Asian | Female | 72 |
| 20–29 | Middle Eastern | Female | 70 |
| 50–120 | East Asian | Female | 69 |
| 40–49 | Indian | Male | 66 |
| 50–120 | Southeast Asian | Female | 58 |
| 40–49 | Indian | Female | 58 |
| 50–120 | Black | Female | 55 |
| 40–49 | Southeast Asian | Male | 54 |
| 40–49 | Southeast Asian | Female | 47 |
| 50–120 | Indian | Male | 41 |
| 30–39 | Middle Eastern | Female | 38 |
| 50–120 | Southeast Asian | Male | 37 |
| 50–120 | Indian | Female | 28 |
| 40–49 | Middle Eastern | Female | 24 |
| 50–120 | Middle Eastern | Female | 16 |

Focus your judgment on the people depicted, ignoring differences in head position/direction, facial expression/emotion, lighting, accessories, and background/objects". Each HIT consisted of 20 unique triplets, where each triplet was presented separately and in succession. We collected 703,300 unique triplet similarity judgments (i.e., one judgment per triplet) on AMT from 1,887 eligible participants. Each triplet was judged once by a single participant, resulting in a single judgment per triplet. To control for quality, according to the criteria outlined in Table 5, we excluded triplet judgments obtained from any participant who provided overly fast, deterministic, or incomplete judgments. This left 638,180 judgments from 1,645 participants. Irrespective of estimated quality, all participants were compensated at a rate of 15 USD per hour. Table 7 shows the age, ancestry, gender identity, and nationality group counts of the 1,645 participants.

**Collection of human behavioral odd-one-out similarity judgments for testing: Same images, novel triplets.** We sampled an additional 1,000 triplets (images were sampled from the stimulus set

Table 7: Age, ancestry, gender identity, and nationality group counts for the 1,645 eligible participants who passed our quality checks and contributed 638,180 triplet judgments

| Age | Count |
|---|---|
| 30–39 | 612 |
| 40–49 | 373 |
| 50–120 | 333 |
| 20–29 | 327 |

| Ancestry | Count |
|---|---|
| Europe | 711 |
| Asia | 331 |
| Americas | 256 |
| Americas and Europe | 175 |
| Africa | 50 |
| Africa and Americas | 21 |
| Asia and Europe | 21 |
| Africa, Americas, and Europe | 20 |
| Africa, Europe | 16 |
| Americas and Asia | 15 |
| Americas, Asia, and Europe | 6 |
| Europe and Oceania | 6 |
| Africa, Americas, Asia, Europe, and Oceania | 5 |
| Africa, Americas, Asia, and Europe | 3 |
| Africa, Asia, and Europe | 3 |
| Africa and Asia | 2 |
| Oceania | 2 |
| Americas, Europe, and Oceania | 1 |
| Asia, Europe, and Oceania | 1 |

| Gender identity | Count |
|---|---|
| Male | 849 |
| Female | 788 |
| Other gender identity | 8 |

| Nationality | Count | Nationality | Count |
|---|---|---|---|
| American | 981 | Thai | 2 |
| Indian | 260 | Greek | 2 |
| Brazilian | 74 | Colombian | 2 |
| Canadian | 44 | Pakistani | 2 |
| Italian | 42 | Bulgarian | 2 |
| English | 39 | Japanese | 1 |
| British | 37 | Puerto Rican | 1 |
| German | 21 | Liberian | 1 |
| French | 15 | Moldovan | 1 |
| Spanish | 13 | South Korean | 1 |
| Australian | 7 | Northern Irish | 1 |
| Romanian | 5 | Serbian | 1 |
| Venezuelan | 5 | Nepalese | 1 |
| Nigerian | 4 | Scottish | 1 |
| Chinese | 4 | Lithuanian | 1 |
| Welsh | 4 | Slovenian | 1 |
| Filipino | 4 | Somali | 1 |
| Armenian | 4 | Russian | 1 |
| Vietnamese | 4 | South African | 1 |
| Polish | 3 | Belizean | 1 |
| Irish | 3 | Sri Lankan | 1 |
| Turkish | 3 | Cameroonian | 1 |
| Portuguese | 3 | Belgian | 1 |
| Dutch | 3 | Trinidadian | 1 |
| Mexican | 3 | Argentine | 1 |
| Ukrainian | 3 | Latvian | 1 |
| Macedonian | 2 | Austrian | 1 |
| Brasileiro | 2 | Icelandic | 1 |
| Estonian | 2 | Vincentian | 1 |
| Jamaican | 2 | Tunisian | 1 |
| Kenyan | 2 | Taiwanese | 1 |
| Jordanian | 2 | Hong Konger | 1 |
| Malaysian | 2 | Ethiopian | 1 |
| Indonesian | 2 | Afghan | 1 |
| Czech | 2 | | |

of 4,921 images). We collected 25 unique judgments per triplet on AMT (total: 25,000 judgments). After excluding responses based on the quality control criteria outlined in Table 5, we were left with 22–25 unique judgments per triplet (total: 24,060 judgments) obtained from 355 participants. Table 8 shows the age, ancestry, gender identity, and nationality group counts of the 355 participants.

**Collection of human behavioral odd-one-out similarity judgments for testing: Novel images, novel triplets.** We sampled all possible triplets from 56 novel images (sampled from FFHQ, but disjoint from the stimulus set) and collected 3 unique judgments per triplet on AMT (total: 83,160 judgments). After excluding responses based on the quality control criteria outlined in Table 5, we were left with 2–3 unique judgments per triplet (total: 80,300 judgments) obtained from 632 participants. Table 9 shows the age, ancestry, gender identity, and nationality group counts of the 632 participants.

**Collection of dimension labels.** For each of the 22 dimensions, we constructed a scale (measurement units corresponded to dimensional values which were converted to percentiles) and tasked participants with providing up to three labels (minimum one label) using words only. Note that participants were not restricted to entering single word labels, they were provided with free-text boxes. Participants were presented with a scale and were asked: "Which *visual* characteristics do you *think* the people at the high end of the scale have in common compared to the people at the low end of the scale?". In addition, participants were asked to view and interact with the entire scale before providing any labels. Figure 5 shows an example of a scale and a novel face image shown to participants. After quality control, there were 25–62 labels per dimension. Quality control roughly entailed removing responses that did not correspond to words and those that were derogatory in nature. The filtered labels were then converted into 35 broad topic categories: "Age", "Age related", "Ancestry", "Ear shape", "Eye color", "Eye related", "Eye shape", "Eyebrow related", "Eyebrow shape", "Face related", "Face shape", "Facial expression", "Facial hair", "Forehead shape", "Gender

Table 8: Age, ancestry, gender identity, and nationality group counts for the 355 eligible participants who passed our quality checks and contributed 24,060 triplet judgments in the "same images, novel triplets" test setting

| Age | Count |
| --- | --- |
| 30–39 | 143 |
| 40–49 | 77 |
| 50–120 | 76 |
| 20–29 | 59 |

| Ancestry | Count |
| --- | --- |
| Europe | 131 |
| Asia | 92 |
| America | 62 |
| America and Europe | 35 |
| Africa and Europe | 8 |
| Africa | 7 |
| America and Asia | 4 |
| Africa, America, and Europe | 4 |
| America, Asia, and Europe | 3 |
| Africa, America, Asia, Europe, and Oceania | 3 |
| Africa and America | 3 |
| Asia and Europe | 2 |
| Africa and Asia | 1 |

| Gender identity | Count |
| --- | --- |
| Male | 184 |
| Female | 166 |
| Other gender identity | 5 |

| Nationality | Count | Nationality | Count |
| --- | --- | --- | --- |
| American | 211 | Venezuelan | 1 |
| Indian | 65 | German | 1 |
| Brazilian | 20 | Swedish | 1 |
| Canadian | 12 | Filipino | 1 |
| Italian | 8 | Estonian | 1 |
| English | 6 | Malaysian | 1 |
| British | 6 | Colombian | 1 |
| French | 4 | Pakistani | 1 |
| Armenian | 3 | Cameroonian | 1 |
| Vietnamese | 2 | Polish | 1 |
| Brasileiro | 2 | Vincentian | 1 |
| Kenyan | 1 | South African | 1 |
| Jamaican | 1 | Thai | 1 |
| Welsh | 1 | | |

expression", "Hair color", "Hair length", "Hair related", "Hair texture", "Hairstyle", "Head shape", "Lip related", "Lip shape", "Mouth related", "Mouth shape", "Neck related", "Nose related", "Nose shape", "Skin color", "Skin related", "Skin texture", "Teeth related", and "Weight related".

Figure 4 provides evidence of the interpretability of the FAX-U (and FAX-C-PH) dimensions. There is clear relationship between the dimension topics obtained from annotator descriptions and dimension labels generated using CelebA and FairFace models.

Table 10 shows the age, ancestry, gender identity, and nationality group counts of the 102 participants.

**Collection of Dimension Ratings.** We tasked participants with placing 20 novel face images along each of the 22 dimensions using the unlabeled scales from the dimension labeling task. We collected 8,800 judgments in total (20 per image-dimension tuple). Participants were presented with a scale and a novel face image above the scale, and were asked: "Decide where on the scale you would place the person shown based on their similarity to the people shown beneath the scale". In addition, participants were asked to view and interact with the entire scale before making their decision. Figure 6 shows an example of a scale and a novel face image shown to participants. Table 11 shows the age, ancestry, gender identity, and nationality group counts of the 164 participants.

# B Implementation Details

## B.1 FAX Models

All images were resized to $128 \times 128$ and normalized to $[-1, 1]$. Standard data augmentation was used (horizontal mirroring and $112 \times 112$ random crops). For the unconditional FAX models, guided by the validation loss, we empirically set $\lambda_1 = 5 \times 10^{-5}$ and $\lambda_2 = 1 \times 10^{-2}$. (The conditional models used the same hyperparameters with the additional loss term's weight set to $\lambda_3 = 1 \times 10^{-4}$.) All models were optimized for $40$ epochs with Adam [37] (default parameters), learning rate $1 \times 10^{-3}$, and batch size 128 on a single Tesla T4 GPU. Post-optimization, we obtained our core set of dimensions by dispensing with those whose maximal value was close to zero, resulting in a low-dimensional space. (For the FAX trained model, this resulted in 22 dimensions.) The threshold for all models was determined based on maximizing accuracy on the validation set. Across five independent runs, the

Table 9: Age, ancestry, gender identity, and nationality group counts for the 632 eligible participants who passed our quality checks and contributed 80,300 triplet judgments in the "novel images, novel triplets" test setting

| Age | Count |
| --- | --- |
| 30–39 | 252 |
| 40–49 | 141 |
| 20–29 | 126 |
| 50–120 | 109 |
| 18–19 | 4 |

| Ancestry | Count |
| --- | --- |
| Europe | 231 |
| Asia | 194 |
| America | 88 |
| America and Europe | 62 |
| Africa | 12 |
| Africa, America and Europe | 10 |
| America and Asia | 8 |
| Africa and Europe | 7 |
| Africa and America | 7 |
| Asia and Europe | 6 |
| America, Asia, and Europe | 2 |
| Africa, America, Asia, and Europe | 2 |
| Asia, Europe, and Oceania | 1 |
| Africa, Asia, and Europe | 1 |
| Africa, America, Asia, Europe, and Oceania | 1 |

| Gender identity | Count |
| --- | --- |
| Male | 350 |
| Female | 279 |
| Other gender identity | 3 |

| Nationality | Count | Nationality | Count |
| --- | --- | --- | --- |
| American | 291 | Mexican | 2 |
| Indian | 164 | Cameroonian | 1 |
| Brazilian | 51 | Chinese | 1 |
| Italian | 19 | Filipino | 1 |
| Canadian | 17 | Colombian | 1 |
| English | 14 | Austrian | 1 |
| British | 10 | Estonian | 1 |
| French | 8 | Jamaican | 1 |
| Spanish | 5 | Indonesian | 1 |
| German | 4 | Irish | 1 |
| Vietnamese | 3 | Japanese | 1 |
| Venezuelan | 3 | Latvian | 1 |
| Kenyan | 3 | Lithuanian | 1 |
| Macedonian | 2 | Malaysian | 1 |
| Greek | 2 | Nigerian | 1 |
| Welsh | 2 | Slovenian | 1 |
| Brasileiro | 2 | South African | 1 |
| Bulgarian | 2 | Sri Lankan | 1 |
| Romanian | 2 | Ukrainian | 1 |
| Thai | 2 | Vincentian | 1 |
| Turkish | 2 | Afghan | 1 |
| Armenian | 2 | | |

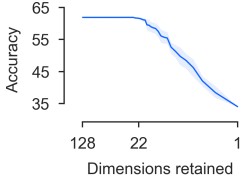 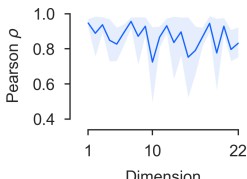 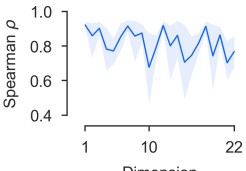

Figure 3: Left: Validation accuracy as a function of the number of dimensions retained over five independent runs (95 % CI), empirically evidencing that the judgments can be encapsulated using a limited number of dimensions. Middle and Right: Pearson and Spearman $r$ between each of the 22 dimensions and the best "matching" dimension from each of the other four runs (95 % CI), empirically evidencing the replicability of dimensions

95% confidence interval (CI) for the number of dimensions and validation accuracy were 27.4±5.8 and 61.9±0.1%, respectively. Figure 3 shows that the judgments can be encapsulated using a limited number of the available dimensions and that the 22 dimensions are approximately replicated across the independent runs.

Note that we discovered some redundancy in the dimensions, in particular 8 dimensions have a Pearson $r > 0.9$ with another dimension. The number of (near) nonzero dimensions depends on the $L1$ sparsity penalty, which must be carefully chosen: Too high of a penalty will result in the entanglement of distinct factors of variation, whereas too low of a penalty will result in repeated dimensions. We erred on the side of caution (lower penalty), since it is more difficult to localize the source of bias when factors are merged, i.e., since our main aim is auditing datasets and models.

### B.2 Baseline models

All images were resized to $128 \times 128$ and normalized to $[-1, 1]$. Standard data augmentation was used (horizontal mirroring and $112 \times 112$ random crops). For preprocessing, we used the same

Table 10: Age, ancestry, gender identity, and nationality group counts for the 102 eligible participants who contributed labels in the dimension naming task

| Age | Count |
| --- | --- |
| 30–39 | 36 |
| 40–49 | 26 |
| 20–29 | 23 |
| 50–120 | 16 |
| 18–19 | 1 |

| Ancestry | Count |
| --- | --- |
| Europe | 40 |
| Asia | 31 |
| America | 14 |
| America and Europe | 9 |
| Africa | 3 |
| Asia and Europe | 1 |
| America and Asia | 1 |
| Africa and Europe | 1 |
| Africa, America, and Europe | 1 |
| Africa and America | 1 |

| Gender identity | Count |
| --- | --- |
| male | 52 |
| female | 50 |

| Nationality | Count | Nationality | Count |
| --- | --- | --- | --- |
| American | 43 | Ethiopian | 1 |
| Indian | 25 | Chinese | 1 |
| Brazilian | 7 | Vietnamese | 1 |
| English | 7 | Venezuelan | 1 |
| Italian | 4 | Icelandic | 1 |
| Spanish | 2 | Lithuanian | 1 |
| Canadian | 2 | Tunisian | 1 |
| German | 2 | Greek | 1 |
| French | 2 | | |

Table 11: Age, ancestry, gender identity, and nationality group counts for the 164 eligible participants who contributed 8,800 triplet judgments in the dimension rating task

| Age | Count |
| --- | --- |
| 30–39 | 55 |
| 20–29 | 37 |
| 40–49 | 36 |
| 50–120 | 35 |
| 18–19 | 1 |

| Ancestry | Count |
| --- | --- |
| Europe | 67 |
| Asia | 40 |
| America | 26 |
| America and Europe | 13 |
| Africa and America | 4 |
| Africa | 4 |
| America, Asia | 3 |
| Africa, America, and Europe | 3 |
| Europe and Oceania | 2 |
| Asia and Europe | 1 |
| Africa and Europe | 1 |

| Gender identity | Count |
| --- | --- |
| Female | 86 |
| Male | 77 |
| Other gender identity | 1 |

| Nationality | Count | Nationality | Count |
| --- | --- | --- | --- |
| American | 70 | Estonian | 1 |
| Indian | 35 | Vietnamese | 1 |
| Brazilian | 12 | Venezuelan | 1 |
| English | 12 | Icelandic | 1 |
| Canadian | 8 | Israeli | 1 |
| British | 4 | Japanese | 1 |
| Italian | 4 | Kenyan | 1 |
| German | 3 | Spanish | 1 |
| French | 3 | Thai | 1 |
| Australian | 1 | Trinidadian | 1 |
| Colombian | 1 | Greek | 1 |

alignment procedure as in [54] to produce $128 \times 128$ face crops. Image values were normalized to $[-1, 1]$.

**CelebA model.** Training was performed using a batch size of 512 (across 4 Tesla T4 GPUs) with SGD for 45 epochs. The initial learning rate 0.1 was divided by 10 at epoch 15, 30, and 40. $L^2$ weight decay and SGD momentum were set to 0.0005 and 0.9, respectively. Standard data augmentation was used (horizontal mirroring and $112 \times 112$ random crops).

Table 12: Predicting human similarity judgments. **(Protocol 1)** Accuracy over 24,060 judgments and Spearman's $r$ between the entropy in human- and model-generated triplet odd-one-out probabilities. **(Protocol 2)** Accuracy over 80,300 judgments and Spearman's $r$ between the entropy in human- and model-generated similarity matrices. Note: #Labels is an approximate of the number of labels collected to create a dataset accounting for consensus.

| Model | Loss/Method | #Images | #Labels | Arch. | Protocol 1 | | Protocol 2 | |
|---|---|---|---|---|---|---|---|---|
| | | | | | Acc. | $r$ | Acc. | $r$ |
| ImageNet-1K | Cross-entropy | 1.28M | >1.28M | RG-32Gf | 46.6 | 0.09 | 41.7 | 0.32 |
| FF | Cross-entropy | 62K | 186K–372K | R18 | 55.9 | 0.41 | 51.9 | 0.67 |
| CA | Cross-entropy | 178K | 7.12M | R18 | 52.1 | 0.25 | 48.9 | 0.56 |
| FFHQ | Cross-entropy | 70K | 140K | R18 | 50.5 | 0.16 | 47.3 | 0.49 |
| CWF | ArcFace | 404K | 404K | R18 | 51.6 | 0.29 | 46.1 | 0.40 |
| CWF | Cross-entropy | 404K | 404K | R18 | 48.6 | 0.21 | 43.2 | 0.34 |
| CWF | SphereFace | 404K | 404K | R18 | 48.3 | 0.25 | 43.8 | 0.35 |
| CWF | CosFace | 404K | 404K | R18 | 44.0 | 0.12 | 40.8 | 0.23 |
| FAX-C | Conditional ?? | 5K | 574K | R18 | *67.4* | *0.68* | *61.7* | 0.82 |
| FAX-C-PH | Conditional ?? | 5K | 574K | R18 | 66.5 | *0.68* | 61.4 | 0.82 |
| FAX-U | Unconditional ?? | 5K | 574K | R18 | 62.0 | 0.65 | 57.5 | *0.86* |
| FAX-U·1/2 | Unconditional ?? | 5K | 287K | R18 | 61.3 | 0.61 | 55.8 | 0.81 |
| FAX-U·1/4 | Unconditional ?? | 5K | 144K | R18 | 60.6 | 0.56 | 55.3 | 0.80 |
| FAX-U·1/8 | Unconditional ?? | 5K | 72K | R18 | 58.6 | 0.47 | 55.0 | 0.79 |
| FAX-Triplet | Triplet margin with distance swap [8] | 5K | 574K | R18 | 60.2 | 0.46 | 52.8 | 0.64 |
| ImageNet-1K | SwAV | 1.28M | 0 | RN50-w5 | 43.8 | 0.08 | 41.0 | 0.30 |
| IG-1B | SwAV | 1B | 0 | RG-32Gf | 47.2 | 0.18 | 44.7 | 0.45 |
| IG-1B | SwAV | 1B | 0 | RG-64Gf | 48.1 | 0.16 | 44.4 | 0.45 |
| IG-1B | SwAV | 1B | 0 | RG-128Gf | 46.8 | 0.15 | 42.8 | 0.40 |
| IG-1B | SwAV | 1B | 0 | RG-256Gf | 47.8 | 0.17 | 43.1 | 0.41 |
| PASS | MoCo-v2 | 1.28M | 0 | R50 | 42.3 | 0.09 | 40.5 | 0.27 |
| PASS | SwAV | 1.28M | 0 | R50 | 42.4 | 0.12 | 40.4 | 0.27 |
| PASS | DINO | 1.28M | 0 | ViTS-16 | 43.2 | 0.10 | 41.8 | 0.32 |

**FairFace model.** Training was performed using a batch size of 512 (across 4 Tesla T4 GPUs) with SGD for 45 epochs. The initial learning rate 0.1 was divided by 10 at epoch 15, 30, and 40. $L^2$ weight decay and SGD momentum were set to 0.0005 and 0.9, respectively. Standard data augmentation was used (horizontal mirroring and $112 \times 112$ random crops).

**CASIA-WebFace models.** We experimented with several face recognition models that differed only in terms of the loss function minimized: Softmax, ArcFace [17], CosFace [76], and SphereFace [45]. Training was performed using a batch size of 512 (across 4 Tesla T4 GPUs) with SGD for 55 epochs. The initial learning rate 0.1 was divided by 10 at epoch 15, 30, and 40. $L^2$ weight decay and SGD momentum were set to 0.0005 and 0.9, respectively. Standard data augmentation was used (horizontal mirroring and $112 \times 112$ random crops).

## C  Additional experiments

**Predicting human similarity judgments.** When appropriate, as baselines, we compare to supervised and self-supervised representation learning methods. Supervised models minimize a cross-entropy, ArcFace [17], CosFace [76], SphereFace [45], or triplet margin with distance swap [8] loss. Self-supervised approaches correspond to MoCo-v2 [15], SwAV [12], or DINO [13]. For data, supervised methods train on CASIA-Webface [82], CelebA [46], FairFace [32], FFHQ [33, 54], or ImageNet-1K [61]. Self-supervised approaches learn on IG-1B [21], ImageNet-1K [61], or PASS [5]. As is standard, baseline representations are extracted from the final encoder layer of a model and then normalized to unit-length. The dot product of two representations determines their similarity. Results for **Protocol 1** and **Protocol 2** are shown in Tbl. 12.

### C.1  Prototypicality

To test whether dimensional values represent the typicality of faces, we use face images from Chicago Face Database (CFD) [49, 40, 50] labeled with *prototypicality* ratings. The ratings (obtained from human annotators) correspond to the average prototypicality of a face wrt a race category from one (less typical) to five (very typical), considering skin color, hair, eyes, nose, cheeks, lips, and other physical features. For gender expression, ratings correspond to the typicality of the face relative

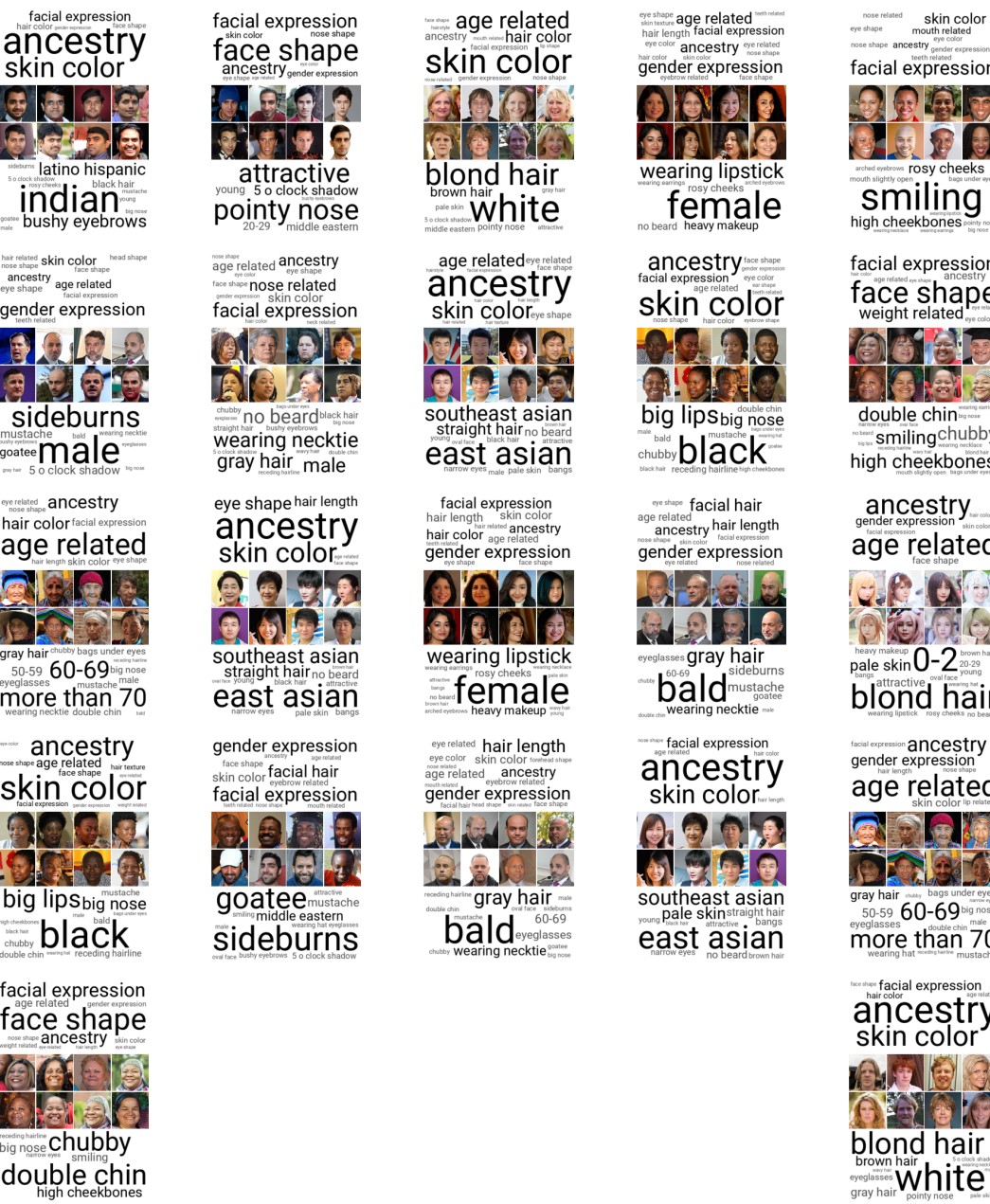

Figure 4: **Dimension labeling.** The top 8 faces from the stimulus set with the highest dimension embedding values for each of the 22 `FAX-U` (and `FAX-C-PH`) dimensions. Word clouds generated from annotator labels (transformed into topics) are shown above each set of 8 faces. Word clouds generated using CelebA and FairFace models are shown below each set of 8 faces. The CelebA and FairFace labels were obtained by labeling the entire stimulus set (using the CelebA and FairFace models) and determining the attributes with the highest AUC.

to others of the same race and gender in the United States from one (not at all typical) to seven (extremely typical).

Tbl. 13 shows that relevant `FAX-U` dimensions are positively correlated with the typicality ratings according to Spearman's $r$. Although the odd-one-task does not require an annotator to explicitly categorize any person, category typicality appears to manifest from the similarity judgments for concepts related to race and gender.

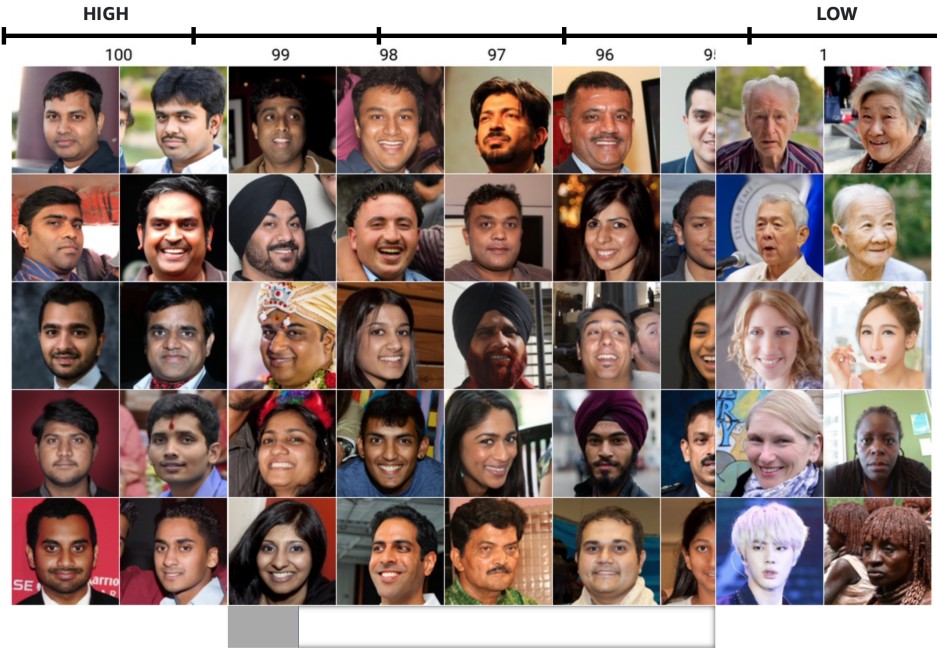

Figure 5: Dimension labeling task. Example scale shown to participants (dimension 1)

Table 13: Prototypicality. Spearman's $r$ between human ratings of face attribute prototypicality and model-generated dimensional values.

| Model | Masculine | Feminine | East Asian | Black | White |
|---|---|---|---|---|---|
| FAX-U | 0.813 | 0.840 | 0.747 | 0.683 | 0.644 |
| CA | 0.858 | 0.855 | — | — | 0.123 |
| FF | 0.836 | 0.827 | 0.806 | 0.653 | 0.645 |
| FFHQ | 0.812 | 0.829 | 0.794 | 0.587 | 0.688 |

**Comparative Dataset Diversity Auditing.** Drawing inspiration from biodiversity measures [42], we propose to use our learned dimensions for comparative dataset diversity auditing. Given a set of candidate datasets $\{c_k\}_{k=1}^n$, we aim to find $c_{k^\star} = \arg\min_{c_k} \operatorname{sim}(c_k)$, where $\operatorname{sim}(c_k)$ measures the similarity of the faces in $x \sim c_k$. We denote by $q_k \in [0, 0.01, \ldots, 0.99, 1]$ and $1 - q_k$ the proportion of images in $c_k$ from $D_0 = \{(x_i, y_0)\}_i$ and $D_1 = \{(x_j, y_1)\}_j$, resp., where $D_0 \cap D_1 = \emptyset$. Here $y_0, y_1 \in \{0, 1\}$ are sensitive attribute labels. Let $\hat{y} \in \mathbb{R}$ denote a *proxy* continuous attribute value. We define the similarity between $(x_i, x_j) \sim c_k$ as $z_{ij} = \operatorname{abs}(\hat{y}_i - \hat{y}_j)^{-1}$. Further, let $Z_k = (z_{ij}) \in \mathbb{R}^{|c_k| \times |c_k|}$ denote $c_k$'s similarity matrix. In biodiversity terms, the average *ordinariness* of faces in $c_k$ is $\propto \sum_i \sum_j z_{ij}$ ($\forall i$). This quantity is large when most faces in $c_k$ are *concentrated* into a few very similar faces. Importantly, concentration is inversely connected to diversity. Therefore, we can interpret the mean of $Z_k$ as a diversity score, which we denote by $\operatorname{divscore}(Z_k)$. Note that $\operatorname{divscore}(Z_k) \to 0$ for homogeneous sets of faces.

**Auditing Model Behavior.** Beyond dataset auditing, as an example, we study the disparate impact of the face image restoration model PULSE [52]. We select PULSE due to its much discussed racial bias [63, 52, 75]. Let $R : x_L \mapsto R(x_L) = x'_H$ denote the PULSE model, which maps a low-resolution face image to a high-resolution face image. We denote by $x_H$ the ground-truth high-resolution face image. Suppose $z_H$ and $z'_H$ denote face attribute representations of $x_H$ and $x'_H$, resp. Disparate outcomes in attribute changes due to $R$ across demographic groups may be an indicator of data and/or algorithmic bias. For face data, we use CFD [49], centering our analysis on self-identified sensitive labels: `Black`, `White`, `Indian`, and `(East) Asian`. We create low-resolution faces by downsampling from $1024 \times 1024$ to $32 \times 32$. For each group, we calculate

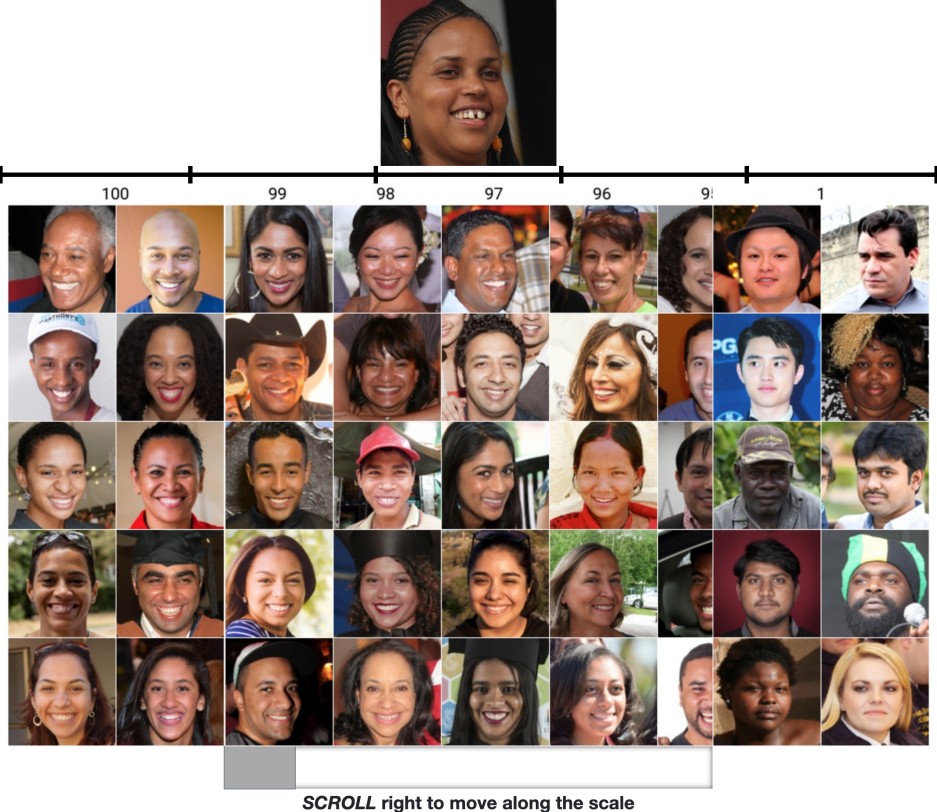

Figure 6: Dimension rating task. Example scale shown to participants (dimension 1)

Table 14: Attribute disparity. Using attribute values from relevant model dimensions to find the most diverse candidate dataset wrt an attribute of interest.

| | | Model (Disparity $\Delta$ / Spearman's $r$) | | | |
|---|---|---|---|---|---|
| Dataset | Attribute | FAX-U | CA | FF | FFHQ |
| CC | $> 70$ y.o.⋆ | 0.22 / 0.96 | *0.06 / 0.99* | *0.06 / 0.95* | 0.08 / *0.99* |
| CC | Male⋆ | 0.06 / 0.97 | *0.02 / 0.99* | *0.02 / 1.00* | 0.04 / 0.99 |
| CC | Light skin | *0.06 / 0.94* | 0.3 / 0.65 | 0.18 / 0.85 | *0.06 / 0.99* |
| CFD | Happy open mouth | *0.12 / 0.99* | 0.06 / 0.95 | — | — |
| CFD | Male⋆ | *0.00 / 1.00* | 0.08 / 0.99 | 0.02 / *1.00* | 0.02 / *1.00* |
| CFD | East Asian⋆ | 0.06 / *1.00* | — | *0.02 / 0.98* | 0.02 / *1.00* |
| CFD | Black⋆ | 0.10 / 0.98 | — | 0.14 / 0.97 | *0.00 / 1.00* |
| CFD | White⋆ | 0.04 / *1.00* | *0.02 / 0.99* | 0.04 / 0.97 | 0.04 / 0.99 |
| CFD | Indian⋆ | *0.08 / 1.00* | — | 0.30 / 0.73 | 0.08 / 0.87 |

the mean cosine similarity between each $(z_H, z_H')$ pair and report the min-max group ratio, i.e., *worst-case* scenario [20]. For the CelebA and FairFace models, attribute representations correspond to unnormalized classifier predictions (logits), whereas for the FAX and CASIA-WebFace-AF models we use the face embeddings.

The min-max ratio for the FAX, CelebA, FairFace, and CASIA-WebFace-AF attribute representations are 0.78, 0.88, 0.58, and 0.98, resp. All attribute representations result in the same min-max pair, i.e., Black-White. Using FAX, CelebA, and FairFace attribute representations, we find statistically significant differences in 22/22, 36/40, and 18/18, resp., for Black individuals, whereas for White individuals, we find statistically significant differences in 16/22, 26/40, and 12/18, resp. All methods show that Black individuals undergo more significant attribute changes. Fig. 7 plots the mean of $z_H' - z_H$, highlighting FAX-model attribute changes per group. To ease presentation, we combined FAX-model dimensions measuring the same attribute. First, Black individuals have highly reduced values in the Black dimension and magnified values in the White dimension, corroborating previous

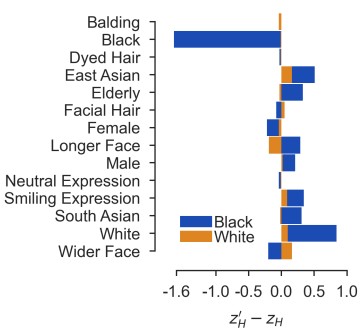

Figure 7: FAX dimensions depicting changes due to the PULSE restoration process on self-identified Black and White groups

qualitative findings [75]. Second, the impact of PULSE on `White` individuals largely centers on changes in face width and length, representing a novel insight. While neither *type* of modification is desirable, the inadvertent erasure of minority groups is extremely harmful. FAX dimensions are therefore useful as a tool for gaining insight into the behavior of face-based models.

