# OpenReview forum: "A View From Somewhere: Human-Centric Face Representations"
_NeurIPS.cc/2022/Workshop/TSRML — TSRML2022_

### Official Review · Reviewer_FK7S · 2022-10-20
**A good paper introducing a novel dataset of human judgments of face similarity**

**Overall Rating:** 6

**Summary:**

This paper introduces a novel dataset, FAX, to evaluate the face attribute diversity in unlabeled datasets. In addition, the authors propose a conditional model that learns to predict human judgments of face similarity. They demonstrate that an annotator’s demographics influences the importance they place on different attributes.

**Strengths:**

- This paper introduces a new public dataset of odd-one-out face similarity judgments. This is important since odd-one-out judgments implicitly encode the dimensions relevant to pairwise similarity which is easier than explicit pairwise similarity.
- The proposed method is evaluated on various face recognition models and datasets.


**Weaknesses:**

- The paper's writing requires further improvement. It is difficult to follow certain sections of the paper. In addition, the conclusion should contain a concise summary of the paper's key points.
- The authors only evaluate their models using ResNet18 backbones and 128-dimensional embeddings.


**Overall Recommendation:**

This paper introduces a new dataset of human judgments of face similarity which will be beneficial to the community. This paper requires additional evaluations and improved writing.

**Review Confidence:**

3: The reviewer is fairly confident that the evaluation is correct

---

### Decision · Program_Chairs · 2022-10-23

Accept